# Scaling Up the Process of Titanium Dioxide Nanotube Synthesis and Its Effect on Photoelectrochemical Properties

**DOI:** 10.3390/ma14195686

**Published:** 2021-09-29

**Authors:** Mariusz Szkoda, Konrad Trzciński, Zuzanna Zarach, Daria Roda, Marcin Łapiński, Andrzej P. Nowak

**Affiliations:** 1Department of Chemistry and Technology of Functional Materials, Faculty of Chemistry, Gdańsk University of Technology, Narutowicza 11/12, 80-233 Gdańsk, Poland; trzcinskikonrad@gmail.com (K.T.); zuzanna.zarach@pg.edu.pl (Z.Z.); dariaroda@gmail.com (D.R.); 2Advanced Materials Center, Gdańsk University of Technology, Narutowicza 11/12, 80-233 Gdańsk, Poland; marcin.lapinski@pg.edu.pl; 3Faculty of Applied Physics and Mathematics, Gdańsk University of Technology, Narutowicza 11/12, 80-233 Gdańsk, Poland

**Keywords:** scaling up, TiO_2_ nanotubes, photoelectrochemical properties

## Abstract

In this work, for the first time, the influence of scaling up the process of titanium dioxide nanotube (TiO_2_NT) synthesis on the photoelectrochemical properties of TiO_2_ nanotubes is presented. Titanium dioxide nanotubes were obtained on substrates of various sizes: 2 × 2, 4 × 4, 5 × 5, 6 × 6, and 8 × 8 cm^2^. The electrode material was characterized using scanning electron microscopy as well as Raman and UV–vis spectroscopy in order to investigate their morphology, crystallinity, and absorbance ability, respectively. The obtained electrodes were used as photoanodes for the photoelectrochemical water splitting. The surface analysis was performed, and photocurrent values were determined depending on their place on the sample. Interestingly, the values of the obtained photocurrent densities in the center of each sample were similar and were about 80 µA·cm^2^. The results of our work show evidence of a significant contribution to wider applications of materials based on TiO_2_ nanotubes not only in photoelectrochemistry but also in medicine, supercapacitors, and sensors.

## 1. Introduction

The advancement of photocatalysis has received a lot of attention throughout the years, with photocatalysis being employed in a considerable range of research areas. When concentrating on photocatalytic processes, TiO_2_ is the most commonly investigated semiconductor, despite the fact that several other semiconductors, namely WO_3_, SnO_2_, or ZnO [1], also offer some advantages for photocatalytic applications. A necessity for effective photocatalysis is that the photocatalyst ought to have a large active surface area. Therefore, nano-sized materials, such as nanoparticles [2], nanotubes [3,4,5], or nanorods [6], have been widely investigated in this respect. The nanotube structure of TiO_2_ has already been used in many applications, including photocatalysis and photoelectrocatalysis [7,8,9,10,11], but also solar cells [12,13,14], electrochromic devices [15,16,17], gas and glucose sensors [18,19,20], as well as medical devices [21,22]. Especially when it comes to photocatalysis, nanosized TiO_2_ is highly regarded due to its photocatalytic water-splitting properties that attract attention as a promising way of hydrogen production. The advantages of low-costs, environmental friendliness, and the possibility of various applications are the cause of the ongoing growth in the interest of nanostructured TiO_2_.

One of the nanostructures that is gaining the most considerable attention is TiO_2_ in the form of nanotubes. A significant amount of research is focused on optimizing the production of these nanostructures and applying appropriate techniques in order to improve their uniformity and control their morphology and their properties. The electrochemical formation of self-organized TiO_2_ nanotubes has become the most common technique; however, other template-assisted methods, sol-gel techniques, and atomic layer depositions are also being used. The influence of several conditions has been discussed, especially during anodic oxidation of the titanium template [23,24,25,26]. For example, Vega et al. combined Laser Interference Lithography with electrochemical anodization and investigated the correlation between the average inter-tube distance and the voltage applied in the anodization process [27]. The results revealed that there is a specific voltage window that provides the synthesis of properly formed nanotubes. Furthermore, one of our previous works [28] was carried out to evaluate the effects of the electrolyte composition on the morphology and properties of the obtained TiO_2_ nanotubes. The experiments focused on the water concentration in the electrolyte bath as well as on the duration of the anodization process and its influence on the overall performance of the material. The conclusion was that the internal diameter of the tubes was increasing along with the ratio of water. Ethylene glycol and the higher contribution of water led to shorter nanotube formations and eventually resulted in the loss of the ordered tubular structure.

Moreover, despite the properties mentioned above, there are only a few studies exploring the possibilities of implementing these processes on a large scale [3,4,29]—most studies report the use of TiO_2_ mainly on a laboratory scale. Furthermore, most studies focus on the implementation of TiO_2_ into the degradation of pollutants in aqueous environments. In the case of the anodization of titanium foil, the lack of reports on the scaling-up process may be caused mainly by obtaining very high values of current, which results in an increased temperature of the electrolyte. As a result, damage to the nanotubular structure may occur. A solution to prevent the destruction of the formed nanostructures is the use of appropriate cooling systems. However, despite widespread knowledge of this solution, the number of reports on this scaling process is still fairly negligible [29]. The synthesis of TiO_2_ nanostructures on larger surfaces makes it possible to obtain a much larger active surface of the material, which is crucial in the case of photocatalytic and photoelectrocatalytic applications.

In this work, we report the facile fabrication of large-scale (up to 64 cm^2^ size) TiO_2_ nanotube layers and their application in photoelectrochemical cells. The samples were studied through scanning electron microscopy (SEM), Raman spectroscopy, and UV–Vis spectroscopy to reveal the impact of scaling up on morphology, crystallinity, and absorbance, respectively. The photoactivity of the obtained titania nanotubes was evaluated according to chronoamperometric measurements performed during exposure to simulated solar radiation. Moreover, the photocurrent values were determined depending on the spot on the sample.

## 2. Materials and Methods

### 2.1. Synthesis of TiO_2_ Nanotubes

Titanium dioxide nanotubes were prepared by anodic oxidation of a titanium foil (99.7%, Sigma Aldrich, Waltham, MA, USA) immersed in a solution consisting of 0.27 M ammonium fluoride, 1 M orthophosphoric acid, 5% *v*/*v* water, and 95% *v*/*v* ethylene glycol. Initially, titanium foils with a thickness of 0.127 mm were cut into pieces with an area of 4, 16, 25, 36, or 64 cm^2^. The given electrode area did not include the additional parts of the plate intended for installation in the measuring system. Afterward, the plates were placed in an ultrasonic bath with a 1:1 mixture of 2-propanol and acetone for 20 min. Clean plates were then electrochemically anodized in a two-electrode system: a titanium plate acted as an anode, while a platinum mesh basket was used as a cathode. Electrochemical anodization was performed for 2 h at a constant voltage of 40 V. During the anodization, the temperature was kept constant at 23 °C using a thermostat (Humber Kiss K6, Huber UK Temperature Control Ltd., Derbyshire, UK). The experimental setup was the same as that proposed in [30]. Throughout the electrolysis, the solution was stirred at 150 rotations per minute (rpm) using a magnetic stirrer (Bionovo, Legnica, Poland). Once the process had been completed, the plates were washed with distilled water and calcinated in the air by heating them to 450 °C for 3 h 45 min; for another 2 h, the temperature was maintained at this level. According to the size of the samples, they were denoted as TN4, TN16, TN25, TN36, and TN64 for areas 4, 16, 25, 36, and 64 cm^2^, respectively. The calcination process was necessary to convert the amorphous phase of the TiO_2_ into a crystal. Figure 1. shows images of the titanium plates after the anodization and calcination processes. 

### 2.2. Characterization Techniques

#### 2.2.1. Morphology and Crystal Structure

The morphology was investigated using scanning electron microscopy (SEM, FEI Quanta FEG 250, FEI Company, Hillsboro, OR, USA). The accelerating voltage was kept at 10 kV. Raman spectra were recorded by a confocal micro-Raman spectrometer (InVia, Renishaw, Wotton-under-Edge, UK) with a sample excitation by means of an argon ion laser, which emitted at 514 nm and operated at 10% of its total power (50 mW). The UV–Vis reflectance spectra of titania nanotubes were measured with a dual-beam UV–Vis spectrophotometer (Lambda 35, Perkin-Elmer, Waltham, MA, USA) which was equipped with a diffuse reflectance accessory. The spectra were registered in the range from 300 to 900 nm, with a scanning speed of 120 nm min^−1^. The crystalline phase of the obtained materials was identified by X-ray diffraction (XRD, Rigaku Miniflex 600, Rigaku, Wilmington, DE, USA) analysis. The electrochemical impedance spectroscopy (EIS) and the Mott–Schottky (MS) plots were carried out on Ivium Vertex potentiostat/galvanostat (Ivium Technology, Eindhoven, The Netherlands). EIS measurements were conducted in a frequency range from 20 kHz to 1 Hz and with 10 mV amplitude. To perform the MS analysis, impedance spectra were recorded at different potentials (potential step 50 mV) between 0.6 V and −0.6 V vs. Ag/AgCl (3 M KCl).

#### 2.2.2. Photoelectrochemical Measurements

In order to analyze the photoelectrochemical properties of the samples, the electrodes were exposed to radiation generated by a 150 W xenon lamp (LOT LS0500/1, LOT, Darmstadt, Germany) equipped with an AM 1.5 filter and a controllable light shutter. The intensity of light was adjusted to 100 mW·cm^−2^ using a Coherent® FieldMate Laser Power Meter. Transient photocurrent studies were performed under a constant polarization of the working electrode, which was set to 0.5 V vs. Ag/AgCl/3 M KCl. Photoelectrochemical measurements were carried out in a 0.2 M K_2_SO_4_ electrolyte. In order to accurately characterize the specific areas of the electrodes, photoelectrochemical tests were carried out in various spots of the samples. For this purpose, several equal fragments of 0.7 cm per 1.5 cm were cut from each TiO_2_NT electrode.

## 3. Results and Discussion

### 3.1. Morphology 

SEM images presenting the morphology of the titania samples and their cross-sections are shown in Figure 2. SEM measurements were performed in the center of each electrode of a different size. All titania samples are composed of regular nanotubes with a similar external diameter (about 100 nm) but slightly varied in length, namely within the range from 1.8 to 2.9 μm. The morphology of the nanotubes obtained on 64 cm^2^ of the titanium foil was slightly different from the others. It can be observed that a greater distance between the nanotubes on the anodized sheets of different areas was measured.

According to the proposed geometric model [31], the real surface area (S_r_) of the titania nanotubes could be estimated based on the following equation:(1)$$S_{r}=\frac{8{\pi \mathrm{hR}}_{2}}{\sqrt{3}{\left(4R_{2}-2R_{1}+y\right)}^{2}}{*S}_{\mathrm{geo}}$$ S_geo_ is the geometric surface area (in this case, it was 1 cm^2^), h is the average length of the nanotube, y is the thickness of the slit between the tubes, R_1_ is the internal radius, and R_2_ is the sum of R_1_ and half of the thickness of the walls of the nanotubes.

The geometrical factors of the formed titanium dioxide (TiO_2_) nanotubes, such as the thickness of the walls of the nanotubes (F), external diameter (D), the length of the nanotubes (H), and the real surface area (S_r_) of the materials are presented in Table 1.

### 3.2. UV–Vis and Raman Spectroscopy

The Raman spectra of the analyzed samples are shown in Figure 3a. The curves were normalized for a clearer presentation of the results. The results of the measurements indicated that the size of the TiO_2_ NT electrode did not change the phase composition of the electrode material in any way. A number of bands characteristics for the anatase crystalline phase were distinguished in the samples. The Raman spectra of TiO_2_ NT exhibits the highest signal at around 143 cm^−1^. Other smaller bands in the range from 400 to 650 cm^−1^ can also be found, which are ascribed to Ti–O stretching vibrations and O–Ti–O symmetric deformation vibrations [32]. 

In Figure 3b, the UV–Vis absorbance spectra for a series of materials are presented (all measurements were made on the center of each sample). As is typical for titanium dioxide, the obtained materials showed a clear absorption edge at about 380 nm in the UV region. Furthermore, a wide absorption band in the visible range of the maximum at 550 nm could be attributed to the presence of sub-bandgap states, which testify to the presence of the structure of the titania nanotubes that can trap light inside of the tubes [33]. The shape of all spectra is similar, but some differences can be observed, particularly in the range from 450 to 650 nm. As can be seen in Figure 2, the TiO_2_ templates varied from each other, mainly due to their different morphology (length and diameter of nanotubes but also the distance between the tubes). Therefore, the emitted light behaved differently in tubes of different shapes, which affected the absorption of visible light within this range.

An XRD analysis was performed to determine the phase composition and crystallinity of the prepared materials (see Appendix A). All detected crystallite phases were indexed as characteristic peaks of anatase and the titanium phase which is derived from the Ti substrate. All patterns were almost identical, which confirms that the size of the anodized sheet does not affect the crystallinity of titanium dioxide. All obtained titanium dioxide nanostructures consisted of the pure anatase phase. This was confirmed by the observed maxima at 25.2°, 38°, 38.2°, 48.1°, 54.2°, and 55.1°, corresponding to the Miller indices (101), (004), (112), (200), (105), and (211), respectively [34].

### 3.3. Photoelectrochemical Properties

In Figure 4, Figure 5 and Figure 6, the transient photocurrent responses recorded at a constant potential of 0.5 V vs. Ag/AgCl/3 M KCl are presented for the analyzed samples. In order to more accurately characterize the electrodes, the photoelectrochemical and SEM (in the center of each cut electrode) measurements were made at various spots on the material. The values of photocurrent density for each sample registered in different places on the electrodes (after 100 s of measurement) are listed in Table 2.

Among almost all samples, depending on the place on which the measurement was made, some differences in photoactivity and morphology were observed. Only in the case of NT4, a comparable value of the generated photocurrents and the shape of nanotubes were obtained. The biggest differences can be seen in the case of the sample obtained during the anodization of the 8 cm × 8 cm sheet. The photocurrent measured at the edges of the electrode was almost four times lower in comparison with the central part of the material. Significant differences can be seen not only in photoactivity but also in the morphology of nanotubes. This is probably due to the fact that the value of the current flowing through the electrode differs depending on the location of the tested electrode fragment; namely, it seems that the current density is higher at the edges of Ti, which contributes to the deformation of the nanotubes. It has already been observed [35] that the layers with no ordered nanotubes were characterized by a lower generation of photocurrents. There is evidence that the non-ordered morphology, characterized by lower electron lifetimes than the nanotubular layer, does not provide optimal pathways for the electrons’ percolation [35,36]. Therefore, it can be concluded that the size of the anodized sheet has a significant impact on the structure of the obtained nanotubes, the ways in which they organize themselves, and their photoactivity. It is worth noting that when the values of the generated photocurrents from the central point of each electrode are compared, they are fairly similar (see Figure 7).

### 3.4. The Solar to Hydrogen Conversion Efficiency 

The solar to hydrogen conversion efficiency (*η*(%)) of the water-splitting reaction was calculated for all the samples using the following equation [37]:(2)$$\eta \left(\%\right)=\frac{J\left(E_{rev}^{0}-(E_{meas}-E_{oc}\right))}{I^{0}}$$
*E_meas_* is the electrode potential (vs. SCE) of the WE at which the photocurrent was measured during illumination, *E_oc_* is the electrode potential (vs. SCE) of the WE under open-circuit conditions, and during illumination, *J* is the photocurrent density, and *I^o^* is the input intensity of the light source. The efficiency calculations were made for all the electrodes and are presented in Figure 8. They are also given in Table 2. As it can be seen, the conversion efficiency values are comparable for all the obtained electrodes (they are about 0.040–0.055%), with one exception for the TN64 material (area 3). The obtained result (as in the case of low photocurrents) is probably due to the lack of an ordered structure in the form of nanotubes.

### 3.5. Flat Band Potential and Carrier Density

To estimate the carrier density (*N_D_*), the Mott–Schottky plots (M–S) of the different electrodes (in a central point) were tested at a frequency of 1000 Hz. The impedance spectra for each electrode, measured at a resting potential without illumination, are presented in the Appendix A. As shown in Appendix A (M–S plots), a positive slope could be observed for all titania samples, which is characteristic of n-type semiconductors and can be determined from its intercept with the E-axis flat band potential (E_fb_). The E_fb_ value is almost identical for each electrode and is around 0.05 V. A value of *N_D_* parameter can be calculated from the following equation:(3)$$N_{D}=\left(\frac{2}{e_{o}}\right){\left[\frac{dC^{-2}}{dE}\right]}^{-1}$$
where: *e* = 1.6 × 10^−19^ C, *ε* = 31 (for anatase) and *ε*_0_ = 8.86 × 10^−14^ F/cm. The corresponding calculated carrier densities of NT4, NT16, NT25, NT36, and NT64 were 1.46 × 10^21^, 2.26 × 10^21^, 1.81 × 10^21^, 9.98 × 10^20^, and 1.09 × 10^21^ cm^−3^, respectively.

## 4. Conclusions

In this study, for the first time, the impact of scaling up the process of titanium dioxide nanotubes and its influence on photoelectrochemical properties were examined. Additionally, the photocurrent density was determined depending on the location of the studied area on the sample. The TiO_2_NTs were prepared by electrochemical anodization using Ti foils with areas of 4, 16, 36, and 64 cm^2^. Morphology studies allowed us to conclude that as the area of the anodized Ti sheets (under the same conditions) increases, the height of the resulting nanotubes decreases and the distance between them grows. However, the wall thickness of the nanotubes remains unchanged. Moreover, it was observed that the changes in their external diameter and the real surface area do not affect the values of the generated photocurrents. The composition of the resulting material also does not change with the scaling-up process. Differences in morphology affect differences in the absorption of visible light in the 450–650 nm range. In general, a larger surface area of nanotubes results in a lower absorbance of light within this range. The effect of the scale on the generation of differences in morphology and generated photocurrents within the same sample was also noted. For the material with an area of 64 cm^2^, the greatest discrepancies were noted among the values of the recorded photocurrents, especially between the central region of the sample and its edges, and the sample exhibited the highest morphology changes regarding deformation of the nanotubes at the edges. It is noteworthy that the photocurrents for all samples, measured at the central point of each surface, had similar values. The size of the sample affects the shape of the nanotubes at the edge of the Ti sheet after anodization. It is evident that the material with an area of four cm^2^ turned out to show the most homogenous morphology among all the samples. The obtained results may have applications in the construction of larger photoelectrochemical systems using TiO_2_ nanotubes but also in supercapacitors or electrochemical sensors.

## Figures and Tables

**Figure 1 materials-14-05686-f001:**
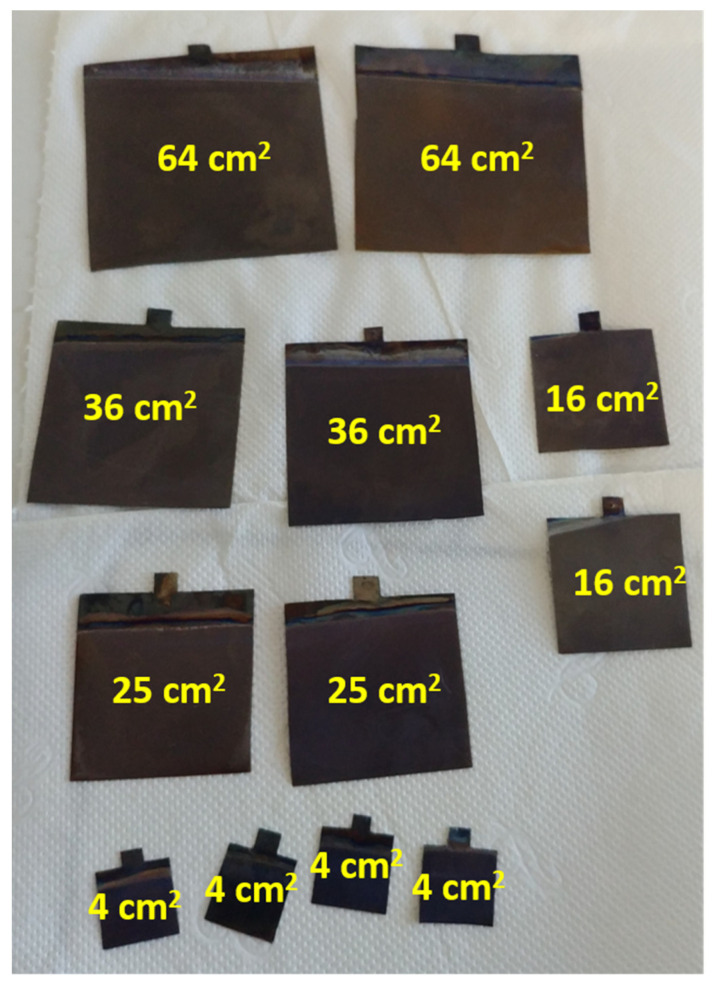
Titanium foil (of different dimensions) after the anodization and calcination process.

**Figure 2 materials-14-05686-f002:**
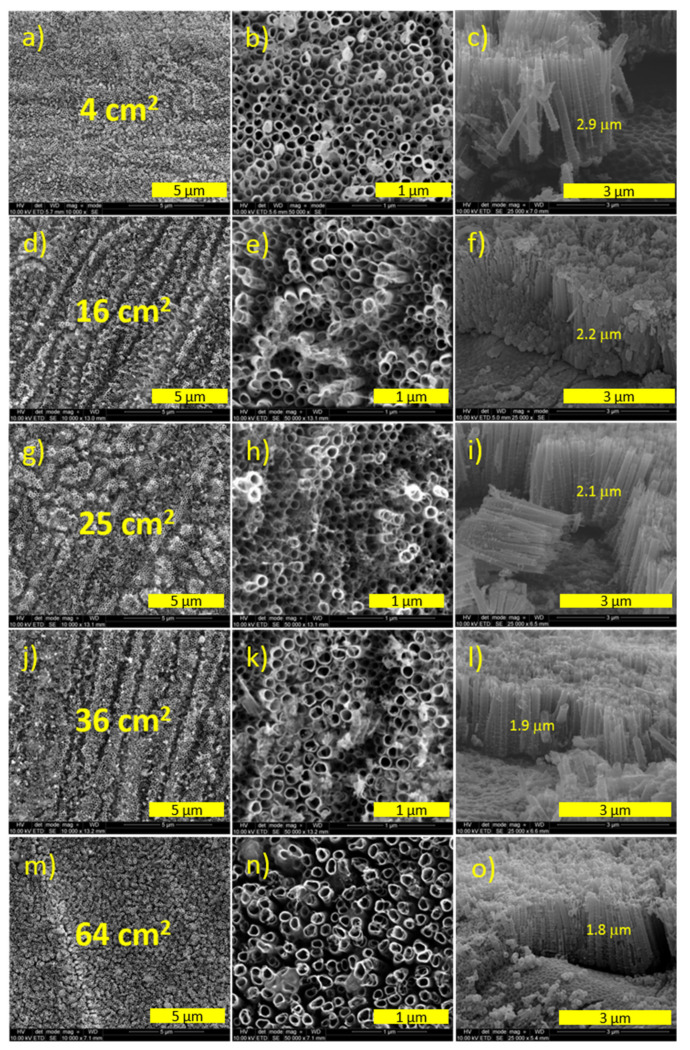
SEM images for TiO_2_ nanotubes depending on size: (**a**–**c**) 4 cm^2^, (**d**–**f**) 16 cm^2^, (**g**–**i**) 25 cm^2^, (**j**–**l**) 36 cm^2^ and (**m**–**o**) 64 cm^2^.

**Figure 3 materials-14-05686-f003:**
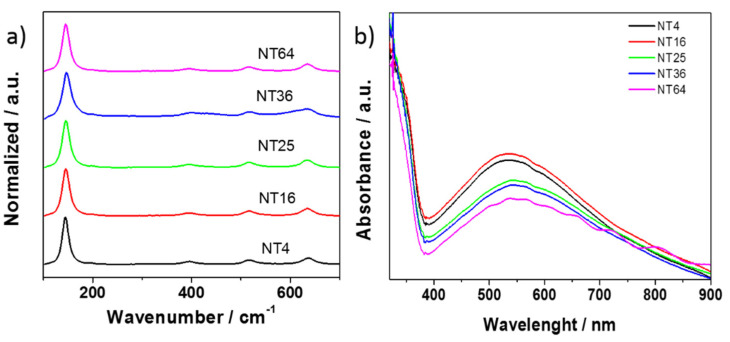
(**a**) Raman spectra and (**b**) the absorbance spectra recorded for a series of titania nanotube electrodes of different dimensions.

**Figure 4 materials-14-05686-f004:**
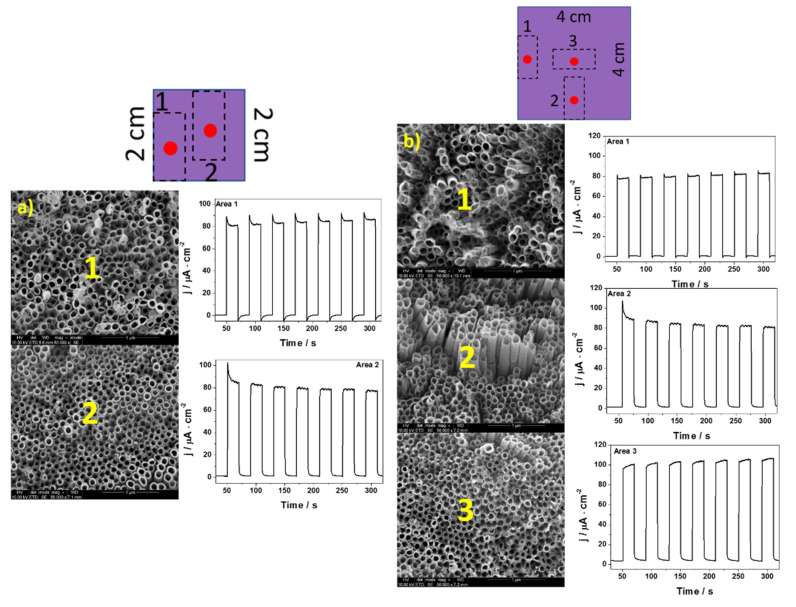
Transient photocurrent responses and SEM at various places on the (**a**) TN4 and (**b**) TN16.

**Figure 5 materials-14-05686-f005:**
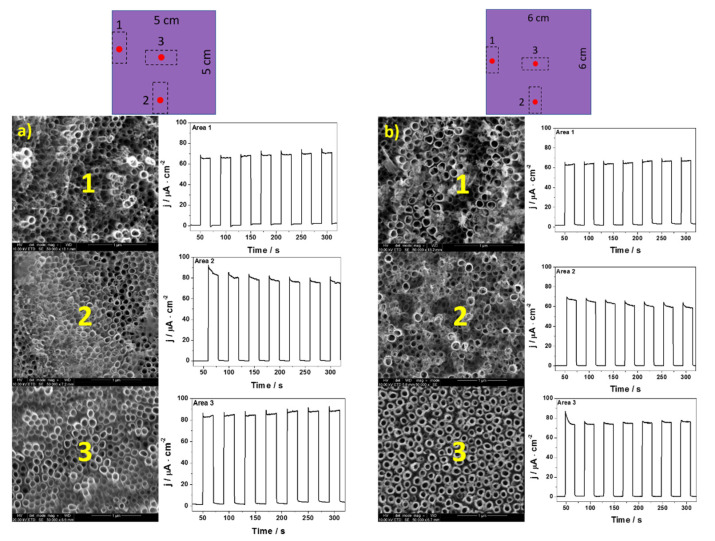
Transient photocurrent responses and SEM at various places on the (**a**) TN25 and (**b**) TN36.

**Figure 6 materials-14-05686-f006:**
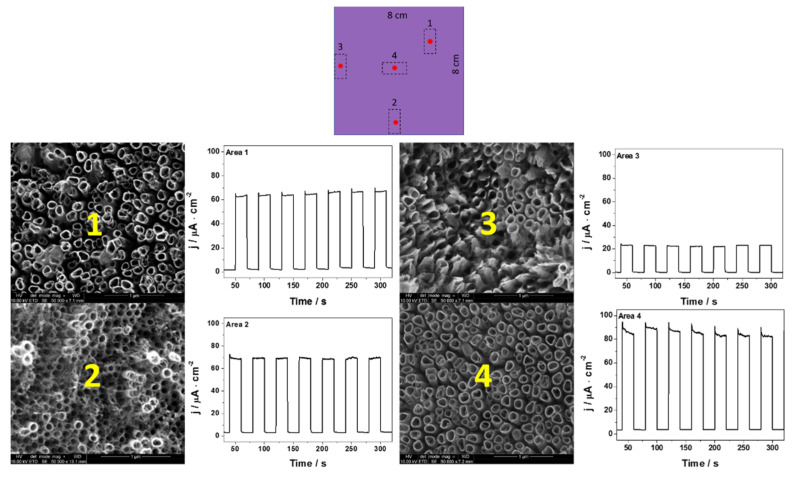
Transient photocurrent responses and SEM at various places on the NT64.

**Figure 7 materials-14-05686-f007:**
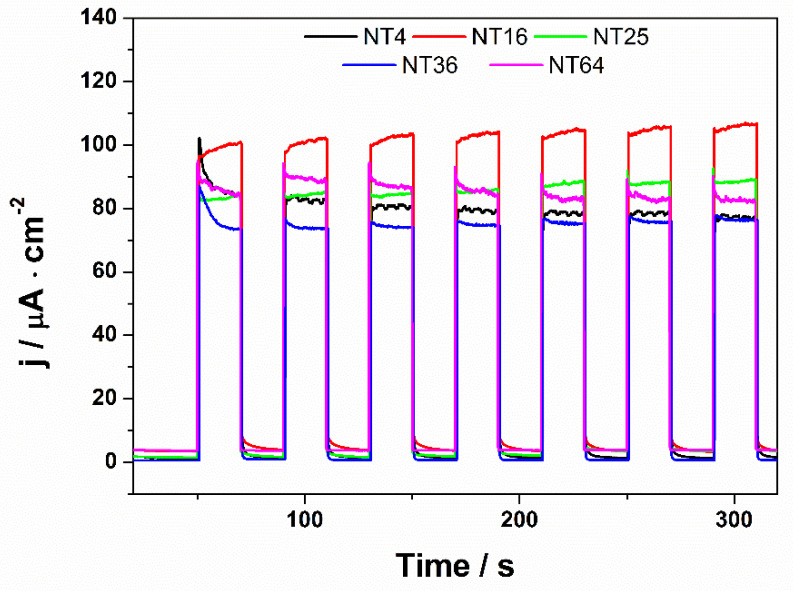
Transient photocurrent response recorded at +0.5 V vs. Ag/AgCl/3 M KCl determined from the central point of each sample.

**Figure 8 materials-14-05686-f008:**
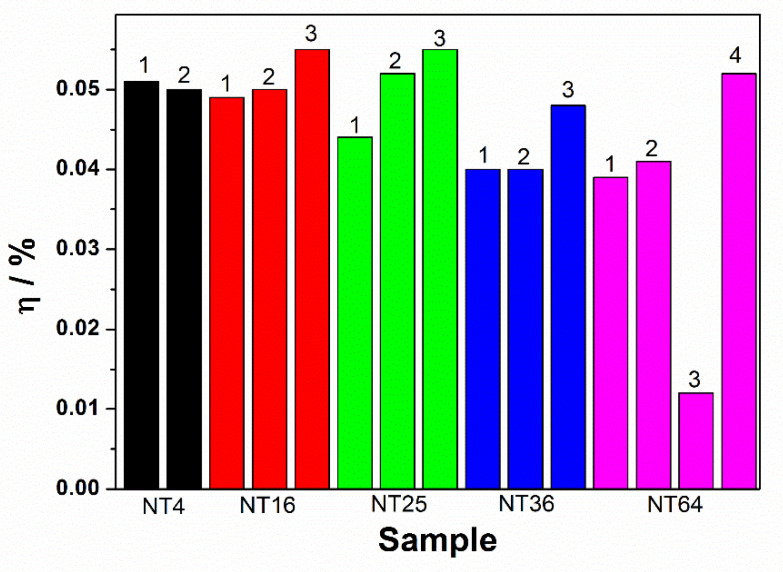
Conversion efficiency values registered for each sample in different places of the electrodes.

**Table 1 materials-14-05686-t001:** The geometrical factors of the titania nanotubes.

No.	Sample	F (nm)	D (nm)	H (µm)	S_r_ (cm^2^)
1	NT4	22	140	2.9	80.8
2	NT16	25	170	2.2	50.3
3	NT25	26	180	2.1	50.2
4	NT36	28	200	1.9	36.8
5	NT64	26	150	1.8	67.1

**Table 2 materials-14-05686-t002:** Density and conversion efficiency values registered for each sample in different places of the electrodes.

Electrode	Area	*J* (µA·cm^−2^)	*η* (%)
NT4	1	83	0.051
	2	81	0.050
NT16	1	80	0.049
	2	82	0.050
	3	99	0.055
NT25	1	66	0.044
	2	79	0.052
	3	82	0.055
NT36	1	62	0.040
	2	63	0.040
	3	75	0.048
NT64	1	62	0.039
	2	66	0.041
	3	22	0.012
	4	83	0.052

## Data Availability

Not applicable.

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
