# Peer review of "Scaling Up the Process of Titanium Dioxide Nanotube Synthesis and Its Effect on Photoelectrochemical Properties"

_materials, 2021, doi:10.3390/ma14195686_

Round 1
Reviewer 1 Report
This work deals with the scaling up of TiO2 nanotubes samples sinthesized by electrochemical anodization technique of Ti foils with different sizes of the areal surface and their photoluminiscence and photoelectrochemical properties. The authors found that by increasing the samples area, the morphology of the TiO2 nanotubes changes by decreasing their size (length), but keeping the same diameter due to the employed anodization voltage of 40 V is the same for all the samples.
There are some queries that should be improved in the manuscript before it can be considered for its publication:
- In the introductory section, the authors shoud compare their statements with the previous ones in the litherature, as the recent works by Vega et al, Electrochimica Acta 203, 51-58 (2016); and Nanoscale Research Letters 2, 355-363 (2007); where the influence of TiO2 nanotube synthesis conditions are discussed.
- The authors do not include the electrolyte they have employed for the anodization of Ti samples in the experimental section.
- Could the authors better expleain why have they have anodized the samples at 23 ºC, and did not carry out the anodization at room teperature instead?
- Experimental bars of the SEM images should be included in the micrographs of TiO2 nanotube samples displayed in the Figure 2 for the different samples size. The figure caption should be properly changed by taking into account the different samples size.
- while the authors have found a reduction in the nanotubes length when increasing the samples areal surface, probably due to the decreasing in the current density during the anodization process, have the authors observed any variation in the nanotubes wall thickness associated to the areal surface of the anodized samples at constant voltage of 40 V? This modification in the nanotubes wall thickness can influence on the photoelectrochemical properties of samples.
Author Response
Reviewer 1
This work deals with the scaling up of TiO2 nanotubes samples sinthesized by electrochemical anodization technique of Ti foils with different sizes of the areal surface and their photoluminiscence and photoelectrochemical properties. The authors found that by increasing the samples area, the morphology of the TiO2 nanotubes changes by decreasing their size (length), but keeping the same diameter due to the employed anodization voltage of 40 V is the same for all the samples.
There are some queries that should be improved in the manuscript before it can be considered for its publication:
Q1. In the introductory section, the authors shoud compare their statements with the previous ones in the litherature, as the recent works by Vega et al, Electrochimica Acta 203, 51-58 (2016); and Nanoscale Research Letters 2, 355-363 (2007); where the influence of TiO2 nanotube synthesis conditions are discussed.
A1. We thank the Reviewer for the suggestion. As requested, the additional text was added to the introductory section. We provide the additional part below:
(…) The advantages of low-cost, environmental friendliness, and the possibility of various applications are the cause of the ongoing growth of the interest of nanostructured TiO2.
One of the nanostructures that is gaining the most considerable attention is TiO2 in a form of nanotubes. A significant number of research work is focused on optimizing the production of these nanostructures and application of appropriate techniques in order to improve the uniformity and control their morphology and thus their properties. The electrochemical formation of self-organized TiO2 nanotubes has become most common technique, however, other template-assisted methods, sol-gel techniques, as well as atomic layer deposition are also being used. Especially during the anodic oxidation of titanium template, the influence of several conditions have been discussed [23–26]. For example, Vega et al. combined Laser Interference Lithography with the electrochemical anodization and they investigated the correlation between the average inter-tube distance and the voltage applied in anodization process [27]. The results revealed that there is a specific voltage window that provides a synthesis of properly formed nanotubes. Furthermore, one of our previous work [28] was carried out to valuate the effect of the electrolyte composition on the morphology and properties of the obtained TiO2 nanotubes. The experiments focused on the water concentration in the electrolyte bath, as well as on the duration of the anodization process and its influence on the overall performance of the material. The conclusion was that the internal diameter of tubes was increasing along with the increasing ratio of water: ethylene glycol, and that the higher contribution of water led to shorter nanotubes formation and eventually resulted in the loss of ordered tubular structure..
Moreover, despite the attractive properties mentioned above, there are only a few studies (…)
Q2. The authors do not include the electrolyte they have employed for the anodization of Ti samples in the experimental section.
A2. Thank you for this comment. In the experimental part, a more detailed description regarding the electrolyte used for the anodization has been added.
Q3. Could the authors better expleain why have they have anodized the samples at 23 ºC, and did not carry out the anodization at room teperature instead?
A3. In our laboratory, the room temperature is set to 23 ºC (air conditioning). During anodization, the temperature of the electrolyte increases due to the large current flow between the electrodes. Therefore, a thermostat was connected to the electrochemical vessel with a jacket cooling in order to maintain the room temperature, i.e. 23 ºC.
Q4. Experimental bars of the SEM images should be included in the micrographs of TiO2 nanotube samples displayed in the Figure 2 for the different samples size. The figure caption should be properly changed by taking into account the different samples size.
A4. As suggested by the Reviewer, both Figure 2. and the caption under the Figure have been changed.
Q5. while the authors have found a reduction in the nanotubes length when increasing the samples areal surface, probably due to the decreasing in the current density during the anodization process, have the authors observed any variation in the nanotubes wall thickness associated to the areal surface of the anodized samples at constant voltage of 40 V? This modification in the nanotubes wall thickness can influence on the photoelectrochemical properties of samples.
A5. Thank you for the important question. As suggested by the Reviewer, nanotubes’ wall thickness was determined and, along with other parameters, presented in Table 1. An additional part added to the manuscript:
According to the proposed geometric model [31], the real surface area (Sr) of the titania nanotubes could be estimated, based on the following equation:
(1)
where Sgeo is the geometric surface area, in this case it was 1 cm2, h is the average length of the nanotube, y is the thickness of the slit between the tubes, R1 is the internal radius and R2 is the sum of R1 and half thickness of the wall of the nanotubes.
The geometrical factors of the formed TiO2 nanotubes such as the thickness of the wall of the nanotubes (F), external diameter (D), the length of the nanotubes (H) and the real surface area (Sr) of materials are presented in Table 1.
Table 1. The geometrical factors of the titania nanotubes.
|
No. |
Sample |
F [nm] |
D [nm] |
H [µm] |
Sr [cm2] |
|
1 |
NT4 |
22 |
140 |
2.9 |
80.8 |
|
2 |
NT16 |
25 |
170 |
2.2 |
50.3 |
|
3 |
NT25 |
26 |
180 |
2.1 |
50.2 |
|
4 |
NT36 |
28 |
200 |
1.9 |
36.8 |
|
5 |
NT64 |
26 |
150 |
1.8 |
67.1 |
As can be seen, the wall thickness is comparable for each material. However, significant differences in the real surface area may have an impact on the photoelectrochemical properties (relevant conclusions are included in the Summary) of the investigated samples.

Reviewer 2 Report
The authors report the facile fabrication of large-scale TiO2 nanotube layers and their application in photoelectrochemical cells.
In general, the presented research is clear and well organized. It is an interesting subject and I suggest accepting it for publishing. There is just one detail that I would suggest to be changed:
Fig 2: please add scaling and magnitude, larger font, on the figures.
Author Response
Reviewer 2
The authors report the facile fabrication of large-scale TiO2 nanotube layers and their application in photoelectrochemical cells.
In general, the presented research is clear and well organized. It is an interesting subject and I suggest accepting it for publishing. There is just one detail that I would suggest to be changed:
Fig 2: please add scaling and magnitude, larger font, on the figures.
Thank you very much for the Reviewer's opinion. As suggested by the Reviewer, both Figure 2. and the caption under the Figure have been changed.

Reviewer 3 Report
Szkoda et al attempted to scale up the synthesis of TiO2 nanotubes by an anodization process. Indeed, they are successful to scale from 4 cm2 to 64 cm2. The study is interesting, therefore, I recommend it for publication after the following comments are addressed.
- The figure quality of 4-6 is very poor, especially the graphical representation of transition photo-current responses. Its hard to read the value son X and Y-axis. It is suggested to improve the figure quality like Figure. 7.
- I understand that this study only focuses on synthesis and scale-up. I suggest deeply investigating for PEC water splitting of the developed materials (photoconversion efficiency, carrier density, etc.)
- Why is the adsorption around 380 nm is not clearly visible in the spectra?
- It is hard to identify the crystallinity of TiO2 nanotubes. XRD measurements are suggested.
- What about the normalized resistance (impedance) offered by TiO2 on scale-up samples? Was it almost the same?
- Why are the TiO2 nanotubes for the 4 cm2 sample is higher in length (2.9 micrometers) compared to other samples?
Author Response
Reviewer 3
Szkoda et al attempted to scale up the synthesis of TiO2 nanotubes by an anodization process. Indeed, they are successful to scale from 4 cm2 to 64 cm2. The study is interesting, therefore, I recommend it for publication after the following comments are addressed.
Q1. The figure quality of 4-6 is very poor, especially the graphical representation of transition photo-current responses. Its hard to read the value son X and Y-axis. It is suggested to improve the figure quality like Figure. 7.
A1. We thank the Reviewer for the comment regarding the visual part of our work. In accordance with the Reviewer’s comment, Figures have been improved.
Q2. I understand that this study only focuses on synthesis and scale-up. I suggest deeply investigating for PEC water splitting of the developed materials (photoconversion efficiency, carrier density, etc.)
A2.We would like to thank the Reviewer for the proposed measurements. As far as possible, we have additionally appointed: the solar to hydrogen conversion efficiency by the water splitting reaction, carrier density and flat-band potential. The text presented was added to the manuscript:
3.4. Efficiency calculation
The solar to hydrogen conversion efficiency (η(%)) of the watert splitting reaction was calculated for all the samples using the following equation [37]:
(2)
where: Emeas is the electrode potential (vs. SCE) of the WE, at which the photocurrent was measured during illumination and Eoc is the electrode potential (vs. SCE) of the WE at open-circuit conditions, also during illumination, J is a photocurrent density, Io an input intensity of a light source. The efficiency calculations were made for all the electrodes and presented in Fig. 8, as well as they are given in Table 2. As can be seen, the conversion efficiency values are comparable for all the obtained electrodes (they are about 0.040-0.055%) with one exception for the TN64 material (area 3). The obtained result (as in the case of low photocurrents) is probably due to the lack of an ordered structure in the form of nanotubes, which is characterized by a lower lifetime of electrons.
Figure 8. Conversion efficiency values registered for each sample in different places of the electrodes.
Table 2. Photocurrent density and conversion efficiency values registered for each sample in different places of the electrodes.
|
Electrode |
Area |
J [µAcm-2] |
|
|
NT4 |
1 |
83 |
0.051 |
|
2 |
81 |
0.050 |
|
|
NT16 |
1 |
80 |
0.049 |
|
2 |
82 |
0.050 |
|
|
3 |
99 |
0.055 |
|
|
NT25 |
1 |
66 |
0.044 |
|
2 |
79 |
0.052 |
|
|
3 |
82 |
0.055 |
|
|
NT36 |
1 |
62 |
0.040 |
|
2 |
63 |
0.040 |
|
|
3 |
75 |
0.048 |
|
|
NT64 |
1 |
62 |
0.039 |
|
2 |
66 |
0.041 |
|
|
3 |
22 |
0.012 |
|
|
4 |
83 |
0.052 |
3.5. Flatband potential and carrier density
To estimate the carrier density (ND), the Mott–Schottky plots of the different electrodes (in a central point) were tested at a frequency of 1000 Hz. The impedance spectra for each electrode, measured at a resting potential, are presented in the Supplementary Information (Fig. S2). As shown in Fig. S3 (M-S plots), a positive slope could be observed for all titania samples, which is characteristic of n-type semiconductors and can be determined from the intercept with E-axis flatband potential (Fb). The Fb value is almost identical for each electrode and it is around 0.05 V. A value of ND parameter can be calculated from the following equation:
(3)
where: e = 1.6·10-19, e = 31 (for anatase) and e0 = 8.86·10-14. The corresponding calculated carrier densities of NT4, NT16, NT25, NT36, and NT64 were 1.46·1021, 2.26·1021, 1.81·1021, 9.98·1020, 1.09·1021 cm-3, respectively.
Q3. Why is the adsorption around 380 nm is not clearly visible in the spectra?
A3. At a wavelength of 380 nm (3.2 eV), the absorption starts to increase (see figure below), what is related to the energy band gap of titanium dioxide (anatase). For higher energies (lower wavelengths) the absorption is visible on the spectrum of each sample. The UV range of the spectrum is characteristic for TiO2, the absorption at the VIS range is related to the morphology of the nanotubes.
Q4. It is hard to identify the crystallinity of TiO2 nanotubes. XRD measurements are suggested.
A4. As suggested by the Reviewer, XRD measurements were performed. A description with a Figure were added to the manuscript. Additional part that was added:
XRD analysis was performed to determine the phase composition and crystallinity of the prepared materials (see Fig. S1 in Supplementary Information). All detected crystallite phases were indexed as characteristic peaks of anatase and the titanium phase which is derived from the Ti substrate. All diagrams are almost identical, which confirms that the size of the anodized sheet does not affect the crystallinity of titanium dioxide. All obtained titanium dioxide nanostructures consisted of the pure anatase phase. This is confirmed by the observed maxima at 25.2°, 38°, 38.2°, 48.1°, 54.2°, and 55.1°, corresponding to the Miller indices (101), (004), (112), (200), (105), and (211), respectively [34].
Figure S1. Comparison of XRD patterns of TiO2 samples.
Q5. What about the normalized resistance (impedance) offered by TiO2 on scale-up samples? Was it almost the same?
A5. Down below we present the results of electrochemical impedance spectroscopy measurements, which were added to the Supplementary Information. The measurements were performed in 0.2M K2SO4 in a frequency range from 20 kHz to 1 Hz, at the resting potential of each electrode. As it can be seen in the spectra, they were almost identical and no significant differences were observed.
Figure S2. EIS spectra of the analyzed samples, performed in 0.2M K2SO4.
Q6. Why are the TiO2 nanotubes for the 4 cm2 sample is higher in length (2.9 micrometers) compared to other samples?
A6. The length of the TiO2 nanotube is different for each sample, not only for 4 cm2. Apart from the length, we have also observed a difference in other parameters of the nanotube, therefore an additional discussion was added to the manuscript:
According to the proposed geometric model [31], the real surface area (Sr) of the titania nanotubes could be estimated, based on the following equation:
(1)
where Sgeo is the geometric surface area, in this case it was 1 cm2, h is the average length of the nanotube, y is the thickness of the slit between the tubes, R1 is the internal radius and R2 is the sum of R1 and half thickness of the wall of the nanotubes.
The geometrical factors of the formed TiO2 nanotubes such as the thickness of the wall of the nanotubes (F), external diameter (D), the length of the nanotubes (h) and the real surface area (Sr) of materials are presented in Table 1.
Table 1. The geometrical factors of the titania nanotubes.
|
No. |
Sample |
F [nm] |
D [nm] |
H [µm] |
Sr [cm2] |
|
1 |
NT4 |
22 |
140 |
2.9 |
80.8 |
|
2 |
NT16 |
25 |
170 |
2.2 |
50.3 |
|
3 |
NT25 |
26 |
180 |
2.1 |
50.2 |
|
4 |
NT36 |
28 |
200 |
1.9 |
36.8 |
|
5 |
NT64 |
26 |
150 |
1.8 |
67.1 |
We believe that a reduction in the nanotubes length when increasing the samples areal surface is probably due to the decreasing in the current density during the anodization process

Round 2
Reviewer 1 Report
The authors have carefully revised and amended the manuscrit by taking into account nearly all queries raised by the referees. There are some minor mistakes and misspellings in the text and references also that should be revised and corrected before the manuscript can be considered for its acceptance for its publication, as e.g, in the reference 23 the authors should be carefully revised to remove some mistakes in their names, and in the reference 27 the list of authors seems not be fully complete. References 28 and 35 are the same and they appear repeated in the list of references.
Reviewer 3 Report
The authors addressed all of my comments very well and I am satisfied with their revisions and therefore, I accept the paper to be published in the present form.